# Identification of Novel Core Genes Involved in Malignant Transformation of Inflamed Colon Tissue Using a Computational Biology Approach and Verification in Murine Models

**DOI:** 10.3390/ijms24054311

**Published:** 2023-02-21

**Authors:** Andrey V. Markov, Innokenty A. Savin, Marina A. Zenkova, Aleksandra V. Sen’kova

**Affiliations:** Institute of Chemical Biology and Fundamental Medicine, Siberian Branch of the Russian Academy of Sciences, Lavrent’ev Ave., 8, 630090 Novosibirsk, Russia

**Keywords:** colitis, colitis-associated cancer, inflammatory bowel disease, colorectal cancer, colon adenocarcinoma, ulcerative colitis, Crohn’s disease, cDNA microarray, transciptomics analysis, microarray

## Abstract

Inflammatory bowel disease (IBD) is a complex and multifactorial systemic disorder of the gastrointestinal tract and is strongly associated with the development of colorectal cancer. Despite extensive studies of IBD pathogenesis, the molecular mechanism of colitis-driven tumorigenesis is not yet fully understood. In the current animal-based study, we report a comprehensive bioinformatics analysis of multiple transcriptomics datasets from the colon tissue of mice with acute colitis and colitis-associated cancer (CAC). We performed intersection of differentially expressed genes (DEGs), their functional annotation, reconstruction, and topology analysis of gene association networks, which, when combined with the text mining approach, revealed that a set of key overexpressed genes involved in the regulation of colitis (*C3*, *Tyrobp*, *Mmp3*, *Mmp9*, *Timp1*) and CAC (*Timp1*, *Adam8*, *Mmp7*, *Mmp13*) occupied hub positions within explored colitis- and CAC-related regulomes. Further validation of obtained data in murine models of dextran sulfate sodium (DSS)-induced colitis and azoxymethane/DSS-stimulated CAC fully confirmed the association of revealed hub genes with inflammatory and malignant lesions of colon tissue and demonstrated that genes encoding matrix metalloproteinases (acute colitis: *Mmp3*, *Mmp9*; CAC: *Mmp7*, *Mmp13*) can be used as a novel prognostic signature for colorectal neoplasia in IBD. Finally, using publicly available transcriptomics data, translational bridge interconnecting of listed colitis/CAC-associated core genes with the pathogenesis of ulcerative colitis, Crohn’s disease, and colorectal cancer in humans was identified. Taken together, a set of key genes playing a core function in colon inflammation and CAC was revealed, which can serve both as promising molecular markers and therapeutic targets to control IBD and IBD-associated colorectal neoplasia.

## 1. Introduction

Colorectal cancer (CRC) is the third most common malignancy and the second leading cause of cancer-related deaths worldwide [1,2]. Colon inflammation, along with the particular host and environmental factors, plays a crucial role in the initiation and progression of CRC [3]. Colitis-associated cancer (CAC) is a type of CRC, which is preceded by clinically detectable inflammatory bowel disease (IBD), including Crohn’s disease (CD) and ulcerative colitis (UC), two highly heterogeneous, incurable, persistent, relapsing/worsening, and immune-arbitrated inflammatory pathologies of the digestive system [4,5]. Epidemiologic studies have showed that patients with IBD have a predisposition to CRC, and cancer risk is highly correlated with the duration and severity of colon inflammation [6,7].

In IBD, chronic long-term colon inflammation accompanied by oxidative stress can alter the expression patterns of key carcinogenesis-associated genes [8]. Moreover, persistent stimulation of epithelial proliferation in the colon by the pro-inflammatory stimuli and excessive cell damage with increased epithelial cell turnover result in detrimental genetic and immunological alterations, making patients with IBD prone to developing CRC [9]. Despite the proven involvement of “inflammation-dysplasia-carcinoma” axis in the malignant transformation of cells in IBD-related CRC [10], the molecular mechanism underlying this process is not yet fully understood. In particular, it remains rather unclear which core genes are involved in the regulation of acute colitis and how markedly their profiles change during colitis-associated malignant transformation of the colon tissue. In addition, the proven complexity of the colitis/CAC-related regulome underlies the low efficacy of conventional IBD/CRC therapy, making it inevitable that surgery is recommended for treating these pathologies [5,11]. Given the known adverse impact of the surgical management of colonic diseases on the quality of life, mental health, and work productivity of patients [5,11], the search for novel key genes involved in the inflammation-related tumor transformation, which can be used as potential molecular targets for IBD therapy, is urgently needed. Moreover, such regulatory genes can be considered as biomarkers of inflammation-driven tumorigenesis and serve as predictors for surveillance strategies and chemoprevention of colitis-related dysplasia and CRC in IBD patients.

To date, extensive exploration of colitis- and CAC-associated regulomes has been performed using transcriptomics-based approaches [12,13,14,15,16,17,18,19,20,21]. Reported bioinformatics studies have revealed some candidate biomarker genes and key signaling pathways susceptible to the development of the mentioned disorders [14,15,16,17,18,19,20,21,22,23], colitis-induced changes in the landscape of immune infiltration of colon tissue [14,16], and a range of hub genes probably involved in the development of CAC [15,20,23]. Despite a plethora of published studies, obtained results are still uncertain and are not well correlated with each other, probably due to insufficient usage of a multiple microarray analysis algorithm (the exploration of three or more independent microarray datasets in the same study, which gives more valid results [18,20,21]), ineffective manual searching of the published literature on the topic of study [14,15,16,17,18,19,20,21], and, in some cases, the absence of proper experimental validation [18]. Since the obtained data still remain insufficient for a thorough understanding of colitis/CAC-associated gene signature, further comprehensive bioinformatics analysis of colitis/CAC-related core genes is required.

In this study, deep re-analysis of multiple microarray datasets related to murine acute colitis (GSE42768, GSE35609, GSE64658, GSE71920, GSE35609) and CAC (GSE31106, GSE5605, GSE64658, GSE42768) was performed. Firstly, the differentially expressed genes (DEGs) were computed between injured and healthy colon tissues, followed by their functional annotation and Venn diagram analysis to identify acute colitis- and CAC-associated core genes. Next, the changes in the sets of core genes associated with the transition from colon inflammation to CRC were identified. Further reconstruction and analysis of gene association networks revealed a range of hub regulators among core genes, subsequent exploration of which by the text mining approach identified a list of candidate genes, which can be used as novel promising biomarkers and therapeutic targets for colitis and CAC. The obtained results were finally validated using an in vivo model of dextran sulfate sodium (DSS)-induced acute colitis and azoxymethane (AOM)/DSS-induced CAC. Furthermore, the role of identified core genes in the colonic carcinogenesis in the backstage of chronic long-term inflammation was analyzed with respect to IBD and CRC in humans.

## 2. Results

### 2.1. Identification of Core Genes Related to Colitis and Colitis-Associated Cancer

To reveal key genes involved in the regulation of acute colitis and its transformation to CAC in mice, a range of independent expression profiles of murine colon tissue were retrieved from the GEO database, including samples of mice of both sexes and different strains with acute colitis stimulated by DSS (GSE42768, GSE35609, GSE64658, GSE71920) or dinitrobenzene sulfonic acid (DNBS) (GSE35609), or chronic colitis driven by azoxymethane (AOM)/DSS accompanied by the development of colorectal cancer (GSE31106, GSE5605, GSE64658, GSE42768). The analysis of selected transcriptomic datasets using the GEO2R tool revealed the sets of differentially expressed genes (DEGs) (colitis vs. control and CAC vs. control) susceptible to the mentioned pathologies, further overlapping of which identified 54 and 109 common DEGs specific to colitis and CAC, respectively (hereafter referred to as core genes) (Figure 1A).

#### 2.1.1. Hierarchical Clustering and Functional Analysis of DEGs

Hierarchical clustering of the expression profiles of identified colitis-associated core genes revealed two main clades separating up- and down-regulated DEGs from each other (Figure 1B). The sub-clade of the most overexpressed DEGs included genes related to immune response (*Ccl3*, *S100a9*, *S100a8*, *Cxcl2*) and heme metabolism (*Hp*), whereas the most suppressed core genes in the colitis group were *Hao2* and *Slc26a3,* associated with fatty acid metabolism and chloride ion transport, respectively (Figure 1B). Further functional analysis of colitis-specific core genes revealed high enrichment of inflammatory-related terms, including the production of pro-inflammatory cytokines IL-1 and TNF-α, IL-17, IGF1-Akt and Tyrobp signaling pathways, antiviral response, matrix metalloproteinases (MMPs), lung fibrosis, and rheumatoid arthritis (Figure 1C, upper panel).

Hierarchical clustering of CAC-specific core genes (Figure 1D) revealed two main clades, grouping activated and suppressed DEGs separately, and one outgroup consisted of the most overexpressed CAC-associated DEGs, notably, regulators of host-microbiota interplay (*Reg3b*, *Reg3g*), immune response (*S100a9*), and extracellular matrix (ECM) remodeling (*Mmp7*). In turn, the most suppressed core genes in the CAC group were involved in the regulation of cell adhesion (*Zan*), pH homeostasis (*Car4*), and ion transport (*Slc26a3*, *Slc37a2, Aqp8*) (Figure 1D). Performed gene set enrichment analysis revealed that CAC-specific core genes are tightly associated with cell invasiveness (wound healing involved in inflammatory response and MMPs), immune response (acute inflammatory response, antimicrobial peptides, etc.), redox imbalance, ion transport, bile secretion, and numerous metabolic processes (Figure 1C, lower panel). Interestingly, the retrieved functional annotation map specific for CAC was significantly less interconnected compared with the acute colitis-associated GO term/pathways network (Figure 1C, upper panel), which can be explained by the more discrete disposition of identified core genes in the CAC-related regulome.

#### 2.1.2. Analysis of Interconnection between Acute Colitis- and CAC-Specific Core Genes

To explore how strongly identified core genes are interconnected in acute and chronic (CAC) phases of colitis, their Venn diagram analysis and the reconstruction of the gene association network were performed. Overlapping of acute colitis- and CAC-related genes demonstrated that 22 of the core genes, playing a regulatory role in acute inflammation, were involved in CAC pathogenesis (Figure 1E), including immune genes (*Ifitm1*, *Ifitm3*, *Il1a*, *Lcn2*, *S100a9*, *Saa3*, *Tnf*), genes encoding protease inhibitors (*Serpina3n*, *Slpi*, *Wfdc18*), ion transporters (*Slc26a2*, *Slc26a3*, *Trpm6*), ECM remodeling proteins (*Mmp10*, *Timp1*, *Mep1a*), signal transduction components (*Igfbp4*, *Lrg1*) and regulators of cell motility (*Capg*), fatty acid homeostasis (*Hao2*), host-microbiota interplay (*Sult1a1*), and heme metabolism (*Hp*).

Analysis of the gene association network generated from acute colitis- and CAC-associated core genes using the STRING database [24] demonstrated their relatively high interconnection: 72 of 141 uploaded core genes (51%) formed interactions with each other within the network (Figure 1F). Interestingly, only 34 of 87 CAC-specific genes (39%) were involved in the network, whereas the shares of acute colitis-specific and common genes in the reconstructed interactome were 66% (21 of 32 genes) and 73% (16 of 22 genes), respectively. Considering that highly interconnected genes can be involved in the same or similar biological processes [25], revealed low enrichment of the analyzed network by CAC-specific genes (Figure 1F) was in line with the discrete structure of the CAC-related functional annotation map shown above (Figure 1C).

Further computing of degree centrality scores of explored core genes revealed a range of genes occupying hub positions in the analyzed network (Figure 1F). It was found that the most interconnected nodes were acute colitis-specific or common genes involved in immune response (*C3*, *Cxcl2*, *Il1b*, *Tnf*) and ECM remodeling (*Mmp9*, *Timp1*). Among CAC-specific genes, the highest degree was identified for stabilizer of endoplasmic reticulum structure *Ckap4*, regulator of cell–cell interaction *Cd44*, and gene *Lyz1* encoding lysozyme (Figure 1F). Given the hub position of *Mmp9* and *Timp1* and the formation of a highly connected cluster of MMPs in the core gene-retrieved network (Figure 1F), the changes in the MMPs profile can be involved in the regulation of malignant transformation of colon tissue during chronic colitis. This pattern needs further clarification.

### 2.2. Identification of Novel Acute Colitis- and CAC-Specific Hub Genes

#### 2.2.1. Acute Colitis-Associated Hub Genes

To identify novel candidate genes for acute colitis and CAC, which can be used as both diagnostic markers and promising therapeutic targets, next we questioned how strongly evaluated core genes can be involved in the regulation of the mentioned pathologies and how well these genes have been studied in the field of inflammatory and neoplastic disorders of the colon.

To address the first issue, the degree centrality scores of the core genes in gene association networks created for each analyzed transcriptomic dataset were computed. Given that hub genes can exert key regulatory functions in reconstructed gene networks [26], the top 20 acute colitis-specific hub genes were identified and are shown in Figure 2A. The obtained results demonstrated that the most interconnected genes associated with acute colitis included genes encoding cytokines (*Tnf*, *Il1a*, *Il1b*), chemokines and its receptors (*Ccl2*, *Cxcl2*, *Ccl3*, *Ccr5*), growth factors and signal transduction components (*Igf1*, *Tyrobp*, *Arrb2*), ECM remodeling regulators (*Mmp3*, *Mmp9*, *Timp1*), and immune (*C3*, *Clec7a*, *H2-Aa*, *Sell*, *Selp*) and protective (*Hp*, *Ugt2b35*) proteins.

Next, to select genes poorly characterized for their role in colitis and colitis-associated disorders, a text mining approach was performed. Analysis of the mention of acute colitis-related core genes (Figure 1B) alongside the keywords “Colitis”, “Crohn’s”, “Dysplasia”, and “Colon cancer” in scientific texts deposited in the MEDLINE database revealed the most studied genes in the field of colitis (*Tnf*, *Il1b*, *Mmp9*, *Igf1*, *Ccl2*, *Slc26a2*, *Timp1*, *Lcn2*, *Il1a*, and *Sell*); the majority of them occupied hub positions in retrieved colitis-associated gene networks (key nodes) (Figure 2A). The rest of the genes were found to be less explored as colitis-related ones (Figure 2B), and, therefore, could be used as a source of novel promising markers/regulators of colitis. To experimentally verify the obtained data, *Mmp3*, *C3*, and *Tyrobp*, displaying, on the one hand, little connection with colitis in the published reports (Figure 2B), and, on the other hand, high degree centrality scores in colitis-associated gene networks (Figure 2A), were selected for further qRT-PCR analysis. Since the profile of MMPs was identified as hypothetically susceptible to transforming acute colitis into CAC (Figure 1F), expressions of *Mmp9* and *Timp1* (known inhibitor of MMPs) were also further validated.

#### 2.2.2. CAC-Associated Hub Genes

The ranking of CAC-specific core genes according to their degree centrality scores in CAC-related gene networks identified the top 20 genes occupying hub positions, including genes encoding cyto- and chemokines (*Tnf*, *Il1a*, *Cxcl16*), regulators of ECM remodeling (*Timp1*, *Mmp7*, *Mmp13*, *Gusb*), immune (*Ctla4*, *Cyba*) and protective (*Gstt1*, *Hp*, *Clu*, *Cyp2s1*) response, lipid homeostasis (*Acss2*, *Chpt1*), ROS production (*Maoa*), cell–cell interaction (*Cd44*), membrane fusion (*Snap25*), and signal transduction (*Plce1*, *Lgr5*) (Figure 2C). Further text mining study, combined with the computing of the association of CAC-related core genes with the overall survival of patients with colon (COAD) and rectal (READ) adenocarcinomas, clearly confirmed the credibility of our bioinformatics analysis: the most reported CAC-related genes (*Tnf*, *Cd44*, *Timp1*, *Mmp7*, *Ctla4*, *Clu*, *Il1a*, *Hp*) were not only associated with poor prognosis in COAD and READ patients but also occupied the hub positions in the networks retrieved from CAC-associated DEGs (Figure 2D). These results indicate a probable important regulatory function of the listed core genes in colitis-associated neoplastic transformation of colon tissue.

To identify novel candidate genes for CAC, our attention was centered on the core genes that are, on the one hand, poorly characterized in the field of CAC, and, on the other hand, associated with ECM remodeling susceptible to “inflammation-dysplasia-carcinoma” axis (Figure 1C,F), notably, *Mmp13* (key node) and *Adam8* (extracellular metalloprotease-disintegrin involved in ECM digestion and markedly associated with pathogenesis of gastrointestinal malignancies [27]) (Figure 2D). In addition, the key nodes *Timp1* and *Mmp7* previously reported as probable regulators of CRC were also selected for qRT-PCR analysis.

### 2.3. Validation of Novel Candidate Genes for Colitis and CAC

#### 2.3.1. Murine Model of DSS-Induced Colitis and CAC

Acute colitis was induced in mice by administration of 2.5% DSS solution in drinking water for 7 days, followed by a 3 day recovery (Figure 3A). CAC was induced in mice by single intraperitoneal (i.p.) injection of AOM 1 week before DSS administration. Furthermore, mice were exposed to 3 consecutive cycles of 1.5% DSS instillations for 7 days, followed by 2 weeks of recovery (Figure 3A). After the experiment termination, the colons were separated from the proximal rectum, mechanically cleaned with saline buffer, and collected for subsequent histological analysis and qRT-PCR.

Gross morphological analysis of healthy colons revealed the normal thickness of the colonic wall and mucosa structure (Figure 3B). Administration of 2.5% DSS for one week led to acute inflammatory changes in the colonic tissues, clearly demonstrating the development of acute colitis and represented by thickening of the colonic wall, hyperemia, hemorrhages, and scattered ulcers (Figure 3B). Long-term cyclic administration of 1.5% DSS with prior injections of carcinogen AOM caused the development of multiple adenomas in the distal part of mice colons with a significant decrease in the intensity of acute inflammatory changes in the colonic tissues (Figure 3B).

Histologically, the colon tissue of healthy mice demonstrated intact colon architecture, non-disrupted crypts, and goblet cells with active mucus vacuoles (Figure 3C). Acute administration of DSS caused severe colon tissue damage, represented by massive epithelium disruption with erosions and ulcerations, diffuse destruction of crypts, and loss of mucosal architecture (Figure 3C). Pronounced inflammatory infiltration through the whole colonic wall, due to neutrophils and lymphocytes as well as mucosa edema, was revealed (Figure 3C). In the case of CAC, chronic administration of DSS after AOM injection caused adenomatous transformation of the colon mucosa, represented by multiple adenomas in the colonic tissue with epithelial hyperproliferation and hyperplastic crypts (Figure 3C). Residual inflammatory infiltration located in the mucosa and submucosa of colon tissue with adenomas and represented by lymphocytes and macrophages was detected (Figure 3C). In the colon tissue adjacent to adenomas (colitis in CAC), signs of chronic colonic inflammation with moderate destruction of the mucosal architecture and crypt damage were found (Figure 3C).

Thus, we reproduced the process of colon carcinogenesis, starting with acute inflammation in the colon tissue, transitioning to chronic inflammation, and eventually ending up with the colonic tumor formation.

#### 2.3.2. Core Genes Expression in the Colonic Tissue of Mice with Acute Colitis and CAC

Finally, the expression of the revealed hub genes related to acute colitis (*C3*, *Tyrobp*, *Mmp3*, *Mmp9*, *Timp1*) and CAC (*Timp1*, *Adam8*, *Mmp7*, *Mmp13*) was validated by qRT-PCR in the colon tissue of mice with acute colitis and colitis-driven adenomas (Figure 3D). As expected, the expression of colitis-related genes *C3*, *Tyrobp*, *Mmp3*, *Mmp9*, and *Timp1* was significantly up-regulated in inflamed colon tissue compared with healthy controls; among them, *Mmp3* and *Timp1* were found to be the most susceptible to acute colitis induction, demonstrating 306.3- and 110.6-fold increases in the expression, respectively, in DSS-treated mice compared with healthy controls (Figure 3D). The chronification of colonic inflammation led to significant reduction in the expression of *C3*, *Tyrobp*, *Mmp3*, and *Timp1* in the adjacent to adenomas colonic tissue by 17.5, 6.6, 2.8 and 46.1 times compared with the samples from acute colitis group, and, moreover, the expression of *C3* and *Tyrobp* in this compartment decreased to the healthy level (Figure 3D). Interestingly, chronification of colitis had no obvious effect on the expression of *Mmp9*: comparable induction of this gene in both DSS- and AOM/DSS-inflamed colon tissues was observed (Figure 3D), which could indicate the important role of *Mmp9* in both acute and chronic colon inflammation, agreeing with [28]. The analysis of colonic adenomatous nodes revealed low expression of all the explored acute colitis-associated key genes: the expression levels of *C3*, *Tyrobp*, *Mmp3*, *Mmp9*, and *Timp1* in adenoma tissue were 26.3, 3.4, 20.8, 11.9, and 27.7 times lower than those in the samples with acute colitis (Figure 3D). Note that adenomatous and adjacent tissues in mice with CAC mainly differed in the expression of the following genes: *Tyrobp* and *Timp1* were found to be 1.9 and 1.7 times overexpressed in adenomas compared with the adjacent counterparts, respectively, whereas *Mmp3* and *Mmp9* were 7.6 and 13.9 times suppressed in tumor tissue, respectively (Figure 3D). Taken together, the obtained results clearly demonstrated that selected key genes associated with acute colitis indeed reached the maximum expression in the acute phase of colon inflammation, whereas chronification of the latter led to a marked decline in this parameter.

As expected, all CAC-associated hub genes (*Adam8*, *Mmp7*, *Mmp13*, and *Timp1* (mentioned above)) were characterized by significant overexpression in tumor nodes compared with healthy tissue, which confirms the expediency of their further exploration as CAC-related marker genes (Figure 3D). Interestingly, only *Mmp7* and *Mmp13* displayed a significantly higher level of activation in colon adenomas compared with both the adjacent tissue (13.8- and 13.4-fold increase, respectively) and colon tissue with acute colitis (186.4- and 19.6-fold increase, respectively). *Adam8* and *Timp1* mentioned above were also up-regulated in adenomas by 8.3 and 1.7 times compared with the adjacent tissue; however, the maximum of their expression was revealed in the acute colitis samples (86.6- and 110.6-fold increase compared with the healthy group, respectively) (Figure 3D).

Thus, the performed qRT-PCR analysis successfully confirmed the expression of the acute colitis- and CAC-related hub genes identified by the in silico analysis in corresponding murine tissues and clearly demonstrated that colitis-driven colonic adenomatous transformation is accompanied by significant changes in the expression profiles of matrix metalloproteinases, which can be used as a novel prognostic signature for colorectal neoplasia in IBD.

## 3. Discussion

Despite the large collection of transcriptomics data from IBD and CAC studies, molecular regulators of the transition of colonic inflammatory lesions to cancer have not yet been clearly defined. The current study aimed to reveal core genes involved in the regulation of acute colitis and CAC development in mice and to explore how far their expression profiles changed during the chronification of colon inflammation.

Performed bioinformatics analysis of multiple cDNA microarray datasets of acute colitis and CAC identified a range of core genes associated with the explored pathologies, further functional annotation of which clearly confirmed the reliability of the obtained data. Indeed, high enrichment of acute colitis-related functional terms with pro-inflammatory cytokines and IGF1-Akt signaling pathway (Figure 1C, upper network) agrees well with the proven regulatory role of the latter in colon inflammation and inflammation-induced mucosal injury [29,30]. Along with this, CAC-related core genes were associated with the processes which markedly changed during colitis-driven tumorigenesis (Figure 1C, lower network): it is known that dysplastic and malignant lesions of colon tissue markedly dysregulate sodium transport [31], bile acid secretion [32], and metabolic [33] and oxidative [34] homeostasis.

Interestingly, the analysis of core genes common for both acute colitis and CAC (Figure 1E) also demonstrated the credibility of the performed in silico study. According to the published reports, the acute-phase genes *Hp*, *Lcn2*, *Lrg1*, and *Serpina3n* included in this list are not only activated in response to inflammatory stimuli, but their aberrant expression is also strongly implicated in tumorigenesis: high levels of *Hp* and *Lcn2* resulted in glucose metabolic dysfunction, angiogenesis, and metastasis in different tumor types [35,36], and *Lrg1* and *Serpina3n* were associated with epithelial–mesenchymal transition in colorectal cancer [37,38]. In addition, the interferon-responsive gene *Ifitm3* is critical to early colon cancer development [12,39], along with *S100a9* and *Slpi*, which, when highly expressed in inflamed colon tissues in mice and patients with colitis and IBD, respectively, can be considered as potent amplifiers of tumor invasion [40,41].

Analysis of gene association networks with subsequent processing of obtained results using the text mining approach revealed a range of core genes occupied hub positions in the acute colitis- and CAC-associated regulomes, which had not yet been extensively studied in relation to the explored diseases (acute colitis: *C3*, *Tyrobp*, *Mmp3*; CAC: *Adam8*, *Mmp13*) (Figure 2). Further qRT-PCR analysis clearly confirmed the overexpression of the mentioned hub genes in the colon tissue of mice with acute colitis and CAC (Figure 3D) that indicated the expediency of further exploration of these genes as promising novel biomarkers of colon inflammation and colon tumorigenesis.

To independently examine how tightly revealed hub genes were associated with inflammation and colorectal cancer, their sub-networks with first gene neighbors from rodent inflammatome [42] and the gene network related to malignant tumors of the colon (DisGeNET ID: C00071202) were reconstructed and analyzed. As depicted in Figure 4, all explored hub genes, except for *Adam8*, indeed form tight modules with gene partners within the evaluated regulomes, and are related to diverse processes and signaling pathways important for the pathogenesis of colitis and CAC. For instance, the detection of the functional group “Interleukin-4 and 13 signaling” is in accordance with [43]: a marked IL-13 response from CD4+ natural killer T cells was previously detected in mice with oxazolone-induced colitis and its blockage was found to ameliorate intestinal inflammation and injury. The members of the integrin family (Figure 4A, Timp1-, C3- and Mmp3-centered sub-networks) play a crucial role in the intestinal homing of immune cells and in supporting the inflammatory mechanisms in the gut [44]. uPA-mediated signaling (Figure 4, Timp1-, Mmp3-, Mmp9-centered sub-networks) controls macrophage phagocytosis in intestinal inflammation, and uPA receptor deficiency leads to marked aggravation of experimental colitis in mice [45]. Moreover, uPA-/- mice demonstrated more severe colorectal neoplasia compared with their wild-type littermates [46]. In addition, remodeling of the extracellular matrix is a hallmark of both colitis/IBD [47] and CAC [48], and prostaglandin signaling is involved in the malignant transformation of inflamed intestinal tissue [49].

The detailed comparison of obtained results revealed a group of MMPs as key participants of acute colon inflammation and its transition to malignancy: functional term “Matrix Metalloproteinases” was identified as statistically significant in both acute colitis- and CAC-associated functional annotation maps (Figure 1C), the highly interconnected cluster of MMPs related to different phases of colitis was revealed in the gene network retrieved from computed core genes (Figure 1F), and MMPs occupied hub positions in all analyzed regulomes related to both acute colitis (Figure 2A: *Mmp3*, *Mmp9*) and CAC (Figure 2B: *Mmp7*, *Mmp13*). Interestingly, the tissue inhibitor of matrix metalloproteinase-1 (*Timp1*) was also detected as a hub gene specific to both acute colitis and CAC (Figure 1E and Figure 2A,B) and tightly interconnected with MMPs module (Figure 1F), which clearly indicated the importance of Timp1/MMPs balance in colitis-induced tumorigenesis. Indeed, Timp1 is a known regulator of colitis, knockout of which markedly attenuated fibrosis in DSS-inflamed colon tissue [50], and, according to the recent report of Niu et al. [51], a hub gene in colorectal cancer regulome. High expression of MMP3 and MMP9 in mucosa-resident macrophages/neutrophils and IgG plasma cells was detected in patients with IBD [52,53]. According to Pedersen et al. [54], MMP3 and MMP9 are two key enzymes involved in the degradation of intestinal tissue during IBD. Interestingly, the silencing of Mmp3 by siRNA markedly ameliorated DSS-induced colitis in mice [55], whereas knockout of Mmp9 or its pharmacological inhibition surprisingly had no obvious effect on the progression of DSS- and TNBS-stimulated colitis in the murine model [56]. Thus, the master regulatory functions of MMPs in colitis pathogenesis require further clarification: in some cases, their overexpression can be considered as a consequence rather than a cause of intestinal inflammation [56]. In the case of CAC-associated MMPs (*Mmp7*, *Mmp13*) revealed in this study, focal high expression of *Mmp7* was previously observed in CAC-related dysplastic lesions [48] and its overexpression was associated with tumor growth, metastasis, and worse overall survival in patients with colon cancer [57]. According to Wernicke et al. [58], the up-regulation of MMP-13 was considered as an early predictive cancer biomarker in patients with colon adenoma, which agrees well with the results of our qRT-PCR analysis (Figure 3D). Despite the extensive studies of MMPs as candidate marker genes of colitis and CAC, to the best of our knowledge, the complex evaluation of the expression of *Mmp3*, *Mmp7*, *Mmp9*, and *Mmp13* in acutely inflamed, adenomatous, and adjacent colon tissues has not yet been reported. Revealed marked changes in their expression profiles during chronification of colitis (Figure 3D) can be considered as a novel gene signature for predicting CAC.

Besides MMPs, another ECM remodeling player, *Adam8*, a member of a disintegrin and metalloproteinase family (ADAMs), was identified as a core gene associated with CAC development (Figure 1F, Figure 2D and Figure 3D). Surprisingly, high expression of *Adam8* was detected not only in CAC but also in DSS-inflamed colon tissue (Figure 3D). Along with the reorganization of ECM, ADAMs are engaged in the processing of various substrates, including cytokines, growth factors, cell adhesion molecules, and receptors, that determines their important role in a range of pathological processes [59]. The most studied ADAMs in IBD was Adam17, associated with EGFR and STAT3 signaling pathways crucial for the pathogenesis of colitis [60], high epithelial expression of which positively correlated with cell proliferation and goblet cell number in UC patients [61]. To the best of our knowledge, the involvement of *Adam8* in the regulation of acute colitis and colitis-induced adenomatous transformation of colon tissue had not yet been reported. Only Christophi et al. and Guo et al. have discussed the overexpression of *Adam8* in IBD patients [62] and AOM/DSS-induced colitis in mice [63]. Given the recently demonstrated ability of Adam8 to control neutrophil transmigration [64] and NLRP3 inflammasome activation [65], the processes tightly associated with colon inflammation [66,67], Adam8 can be considered as a novel promising master regulator of colitis and CAC; this requires further clarification. Interestingly, despite the revealed low interconnection of Adam8 with the colon cancer-associated gene network retrieved from DisGeNET (Figure 4), this gene seems to play an important role in the pathogenesis of CAC: Adam8 is involved in the activation of integrin, FAK, ERK1/2, and Akt/PKB signaling pathways related to cancer progression [68], its overexpression was identified in colorectal cancer compared with adjacent normal tissues [69], and the suppression of the expression of *Adam8* by knockout or siRNA approaches resulted in reduced proliferation and invasiveness of colon cancer cells [69,70].

Finally, *C3* and *Tyrobp* were also revealed as colitis-specific hub genes (Figure 2A and Figure 3D), which is in line with published reports. Previously, a high level of C3 in the serum and jejunal secretion of IBD patients was identified [71,72]. Moreover, C3 was found to be up-regulated in intestinal epithelial cells in the DSS-induced colitis model [73], and its ablation promoted inflammatory responses in the mid colon [74] and significantly reinforced DSS-induced colitis in C3 knockout mice compared with wild-type littermates [72]. Tyrobp is a known regulator of the production of pro-inflammatory mediators in macrophages and neutrophils [75], and, thus, is implicated in pathogenesis of various inflammation-associated diseases [75,76,77]. According to recent studies, Tyrobp was identified as a probable upstream regulator of UC [78], and its knockout robustly attenuated the severity of DSS-induced colitis in mice, whereas its overexpression resulted in a striking exacerbation of colon damage caused by DSS [79].

The published works discussed above demonstrated the involvement of the revealed core genes in the regulation of inflammation and malignant lesion of the colon, not only in murine models but also in patients. To independently confirm the translational bridge between our findings and the pathogenesis of colitis/CAC in humans, expression of core genes (acute colitis: *C3*, *Tyrobp*, *Mmp3*, *Mmp9*, *Timp1*; CAC: *Timp1*, *Mmp7*, *Mmp13*, *Adam8*) was further evaluated in the transcriptomics profiles of colon tissue from patients with UC and CD collected from GEO (Figure 5A) and colorectal cancer retrieved from The Cancer Genome Atlas (TCGA) (Figure 5B). As depicted in Figure 5A, the majority of the explored key genes were overexpressed in IBD and demonstrated more pronounced susceptibility to the induction of UC compared with CD, except for *TYROBP*, expression of which was more up-regulated in CD patients. Interestingly, despite the proven association with CAC (Figure 2B,D), *TIMP1*, *MMP7*, and *ADAM8* were activated in IBD-affected colon tissues (Figure 5A), which is fully in line with our data: the high expression of these genes was demonstrated in DSS-inflamed and adjacent to adenomas colon tissues in mice (Figure 3D). In addition, similar to our results (Figure 3D), CAC-specific MMP13 was found to be slightly associated with IBD: its low activation in two of the four analyzed UC transcriptomics datasets and unchanged levels in CD samples were observed (Figure 5A). Presumably, Mmp13 plays a minor role in ECM remodeling in colitis, whereas CAC was associated with significant up-regulation of its expression, which makes *Mmp13* a promising gene candidate for the predicting of colitis-associated tumorigenesis; this requires further detailed study. TCGA analysis of the identified CAC-related core genes revealed a significant association between high expression of *TIMP1* and *ADAM8* with low overall survival of patients with both colon (COAD) and rectal (READ) adenocarcinomas (Figure 5B). Despite the finding that Timp1 and Adam8 can play important regulatory functions in CAC, this supposition requires further detailed confirmation, since TCGA analysis was performed without consideration of the ratio of UC- and CD-associated CAC patients in COAD and READ cohorts. In addition, given recently reported sex disparities in the association of Timp1 expression with cancer progression [80], further exploration of its regulatory role in CAC in mice of both sexes is needed.

The obtained results were finally summarized in the scheme depicted in Figure 5C. According to our findings, (a) revealed core genes not only occupy hub positions within explored acute colitis- and CAC-specific regulomes, but also are interconnected with each other, (b) *Timp1* is identified as a hub node in gene association networks retrieved for both acute colitis and CAC, which can indicate its crucial role in colitis-associated tumorigenesis, (c) chronification of colonic inflammation is accompanied by a switch in MMPs profile (acute colitis: *Mmp3*, *Mmp9*; CAC: *Mmp7*, *Mmp13*), which can serve as a gene signature panel for prognosis of malignant transformation of inflamed colon tissue; and (d) identified core genes are overexpressed in the colon tissue of patients with IBD (all explored genes) and highly aggressive colorectal cancer (*TIMP1*, *ADAM8*), confirming the interest in studying these genes within the framework of intestinal pathologies in humans (Figure 5C).

### Limitations of the Study

The limitations of the study are as follows: First, given the relatively low number of mice used for experimental validation of the obtained data (*n* = 6), and their belonging to only one sex (female) and one strain (C57Bl6), further study is required to validate the results using a larger sample size obtained from mice of both sexes and different strains. Second, considering that our findings are predominantly animal-based, to more clearly elucidate how closely (if at all) the identified core genes are involved in the regulation of intestinal pathologies in humans, revealed translational bridge needs further large-scale verification study, using clinical samples of patients with UC, CD, and UC/CD-associated colorectal cancer. Third, despite the identification of high degree centrality scores of the explored key genes and their tight association with crucial colitis/CAC-related signaling pathways, the master regulatory functions of these genes in colitis and CAC should be further verified experimentally (for instance, using knockout models).

## 4. Materials and Methods

### 4.1. Microarray Data Collection and Differential Expression Analysis

The gene expression profiles associated with murine acute colitis and CAC, as well as ulcerative colitis and Crohn’s disease, in patients were acquired from the Gene Expression Omnibus database [81] (Table 1). The fold changes between the mean expression values of the genes in the experimental (pathology) versus control groups were computed using the GEO2R tool [82]. The Benjamini–Hochberg false discovery rate method was selected for adjusting *p*-values. The genes with a *p*-value < 0.05 and |fold change| > 1.5 were identified as differentially expressed genes (DEGs) and were collected for further analysis. Overlapping of the DEGs from different datasets was performed using the InteractiVenn tool [83]. Hierarchical clustering of DEGs according to their expression profiles was carried out using the Euclidean distance metric, using the Morpheus tool (https://software.broadinstitute.org/morpheus, accessed on 12 December 2022).

### 4.2. Functional Analysis of DEGs

Functional annotation of acute colitis- and CAC-associated DEGs was performed using the ClueGO 2.5.7 plugin in Cytoscape 3.7.2, using the latest updates of Gene Ontology (Biological Processes), Kyoto Encyclopedia of Genes and Genomes (KEGG), WikiPathways, and REACTOME databases. The GO Tree interval was ranged from 3 to 8 and the minimum number of genes per cluster was set to 3. Enrichment of functional terms was tested using the two-sided hypergeometric test corrected using the Bonferroni method, followed by selecting significantly enriched terms with a *p*-value < 0.05. To cluster similar functional groups retrieved from different databases in the common pathway-specific modules, the GO Term Fusion was used. Functional grouping of finally selected functional terms was performed using kappa statistics (kappa score ≥ 0.4). Functional annotation of gene modules, consisting of core genes and their first gene partners extracted from murine inflammatome and colon cancer-related regulome, was performed using the ToppFun tool (databases: KEGG, REACTOME, MSigDB C2 BIOCARTA, BioSystems: Pathway Interaction Database, Pathway Ontology; Bonferroni adjustment) [84].

### 4.3. Reconstruction of Gene Association Networks

Gene association networks were reconstructed from the genes of interest using the Search Tool for the Retrieval of Interaction Genes (STRING) database, using the stringApp 1.5.1 tool [85], and were visualized using Cytoscape 3.7.2. The cutoff criterion of the confidence score was set as >0.7 to eliminate inconsistent “gene–gene” pairs from the dataset. The number of neighbors of a gene of interest within reconstructed networks was calculated using the NetworkAnalyzer plugin [86] and visualized using the Morpheus platform [87].

### 4.4. Data Mining Analysis

The search for the co-occurrence of the names of core genes with various colitis- and CAC-related terms in the same sentences in abstracts of published reports deposited in the MEDLINE database was performed using the GenCLiP3 tool [88], with the following settings: impact factor of 0–50 and year of publication of 1992–2022. The results were visualized using Circos [89].

### 4.5. Murine Models of Acute Colitis and Colitis-Associated Cancer (CAC)

Eight-week-old female C57Bl6 mice with an average weight of 22–24 g were obtained from the Vivarium of the Institute of Chemical Biology and Fundamental Medicine SB RAS (Novosibirsk, Russia). Mice were housed in plastic cages (7 animals per cage) under normal daylight conditions. Water and food were provided ad libitum. Experiments were carried out in accordance with the European Communities Council Directive 86/609/CEE. The experimental protocols were approved by the Committee on the Ethics of Animal Experiments at the Institute of Cytology and Genetics SB RAS (Novosibirsk, Russia) (protocol No. 56 from 10 August 2019).

Acute colitis was induced in mice (*n* = 10) by administration of 2.5% DSS solution in drinking water for 7 days, followed by 3 days of recovery. Mice were sacrificed on day 10 after colitis initiation. CAC was induced in mice (*n* = 10) by a single intraperitoneal (i.p.) injection of carcinogen AOM (10 mg/kg) 1 week before DSS administration, as described in [90]. Furthermore, mice were exposed to 3 consecutive cycles of 1.5% DSS instillations with drinking water for 7 days, followed by 2 weeks of recovery. The mice were sacrificed 10 weeks after the start of the experiment. At the end of the study, the colons were separated from the proximal rectum, mechanically cleaned with saline buffer, and were then collected. Only 8 of 10 samples had well-formed adenomas in the colon, which were selected for the subsequent gross examination, histological analysis, and qRT–PCR.

### 4.6. Histology

For the histological study, colon specimens were fixed in 10% neutral-buffered formalin (BioVitrum, Moscow, Russia), dehydrated in ascending ethanol and xylols, and embedded in HISTOMIX paraffin (BioVitrum, Moscow, Russia). The paraffin sections (5 μm) were sliced on a Microm HM 355S microtome (Thermo Fisher Scientific, Waltham, MA, USA) and stained with haematoxylin and eosin. The images were examined and scanned using an Axiostar Plus microscope equipped with an Axiocam MRc5 digital camera (Zeiss, Oberkochen, Germany) at magnifications of ×100.

### 4.7. Quantitative Real-Time PCR (qRT-PCR)

Total RNA was isolated from the colons of experimental animals using TRIzol reagent (Ambion, Austin, TX, USA) according to the manufacturer’s instructions. Briefly, colon tissue was collected in 1.5 mL capped tubes, filled with 1 g of lysing matrix D (MP Biomedicals, Irvine, CA, USA) and 1 mL of TRIzol reagent, then homogenized using a FastPrep-24 TM 5G homogenizer (MP Biomedicals, Irvine, CA, USA) with QuickPrep 24 adapter. The homogenization was performed at 6.0 m/s for 40 s. After homogenization, the content of the tubes was transferred to the new 1.5 mL tubes without lysing matrix. Total RNA extraction was performed according to the TRIzol reagent protocol.

Due to the known ability of DSS to linger in the RNA extracted from the colon tissue, and, thus, interfere with both reverse transcription and PCR reactions, the extracted total RNA was diluted to a volume of 250 μL and purified using Microcon Centrifugal Filter Devices (MilliPore, Burlington, MA, USA) by centrifuging for 1 h at 14,000× *g*.

The first strand of cDNA was synthesized from total RNA (*n* = 6 per group, the samples with the highest RNA purity and integrity) in 100 μL of reaction mixture containing 2.5 μg of total RNA, 20 μL of 5× RT buffer (Biolabmix, Novosibirsk, Russia), 250 U of M-MuLV-RH revertase (Biolabmix, Novosibirsk, Russia), and 100 μM of dT(15) diluted to a volume of 100 μL. Reverse transcription was performed at 25 °C for 10 min followed by the incubation at 42 °C for 60 min with subsequent termination at 70 °C for 10 min.

Amplification of cDNA was performed in a 25 μL PCR reaction mixture containing 5 μL of cDNA, 12.5 μL of HS-qPCR (2×) master mix (Biolabmix, Novosibirsk, Russia), 0.25 μM each of the forward and reverse primers to *Hprt* and *Hprt* specific ROX-labeled probe, 0.25 μM each of the forward and reverse gene-specific primers, and FAM-labeled probe (Table 2). Amplification was performed as follows: (1) 94 °C, 2 min; (2) 94 °C, 10 s; (3) 60 °C, 30 s (steps 2–3: 50 cycles). The relative level of gene expression was normalized to the level of *Hprt* expression according to the ΔΔCt method.

Amplification was performed using a C1000 Touch with CFX96 module Real-Time system (BioRad, Hercules, CA, USA), and the relative level of gene expression was calculated using BioRad CFX manager software (BioRad, Hercules, CA, USA). Three to five samples from each experimental group were analyzed in triplicate. The sequences of the primers used in the study are listed in Table 2.

### 4.8. The Association of DEGs Expression with Survival Rates of Patients with Colorectal Cancer

To explore the association of revealed core genes with the progression of colon (COAD) and rectal (READ) adenocarcinomas, analysis of the survival rates and their correlation with the expression of studied genes was performed using The Cancer Genome Atlas (TCGA) clinical data for patients with COAD and READ. Kaplan–Meier survival curves for COAD and READ patients depending on the mRNA expression level of core genes were constructed using the OncoLnc tool [91].

### 4.9. Statistical Analysis

The statistical analysis was performed using Benjamini–Hochberg false discovery rate method (identification of DEGs; GEO2R tool), two-sided hypergeometric test with Bonferroni correction (functional analysis of DEGs; ClueGO plugin and ToppFunn tool), and two-tailed unpaired Student’s *t*-test (qRT-PCR analysis; Microsoft Excel). *p*-values of less than 0.05 were considered statistically significant.

## 5. Conclusions

In summary, this animal-based research revealed a range of core genes associated with acute colitis (*C3*, *Tyrobp*, *Mmp3*, *Mmp9*, *Timp1*) and CAC (*Timp1*, *Mmp7*, *Mmp13*) in mice. The observed high rate of interconnection of these genes with gene networks retrieved for intestinal inflammation and malignancy, their significant association with key colitis/CAC-related signaling pathways, and probable involvement in the pathogenesis of IBD and colorectal cancer in patients demonstrated the expediency of further detailed studies of identified core genes as novel master regulators and promising therapeutic targets for colitis and CAC.

## Figures and Tables

**Figure 1 ijms-24-04311-f001:**
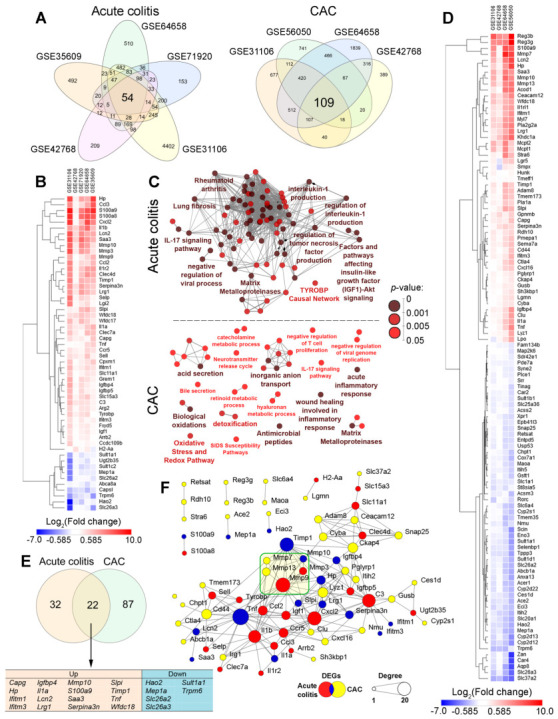
Core genes involved in the development of acute colitis and CAC in mice revealed by bioinformatics analysis. (**A**) Venn diagrams overlapping differentially expressed genes (DEGs) identified by re-analysis of cDNA microarray datasets of colon tissue of mice with colitis and CAC. (**B**) Heatmap demonstrating expression levels of DEGs in acute colitis-related GSE datasets (acute colitis vs. healthy control). (**C**) Functional analysis of overlapped DEGs associated with acute colitis and CAC. Enrichment for Gene Ontology (Biological Processes), KEGG, REACTOME, and WikiPathways terms performed using ClueGO plugin in Cytoscape. Only pathways with *p* < 0.05 after Bonferroni correction for multiple testing were included in the functional annotation map. (**D**) Heatmap demonstrating expression levels of DEGs in CAC-related GSE datasets (CAC vs. healthy control). (**E**) Venn diagrams overlapping revealed acute colitis- and CAC-associated core genes. (**F**) Gene association network retrieved from identified core genes associated with acute colitis and CAC reconstructed using STRING database (confidence score ≥ 0.7). Degree—the number of first neighbors (gene partners) within the gene network.

**Figure 2 ijms-24-04311-f002:**
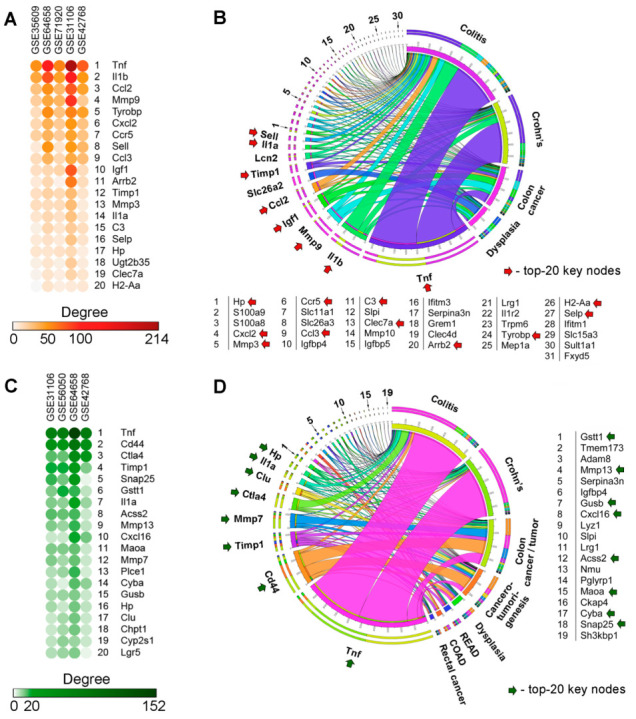
Core genes characterized by hub position within acute colitis- and CAC-related regulomes. (**A**,**C**) Heatmaps demonstrating interconnection of core genes in gene association networks reconstructed for each acute colitis- (**A**) and CAC-related (**C**) dataset using the STRING database (confidence score ≥ 0.7). Only the top 20 of the most interconnected hub genes are represented. (**B**,**D**) Co-occurrence of revealed hub genes with keywords associated with colitis (**B**) and CAC (**D**) in the scientific literature deposited in the MEDLINE database. Analysis was performed using the GenClip3 tool. Data were visualized using Circos. COAD and READ-association of the aberrant expression of hub genes with poor prognosis in patients with colon and rectal adenocarcinomas, respectively, according to the TCGA database.

**Figure 3 ijms-24-04311-f003:**
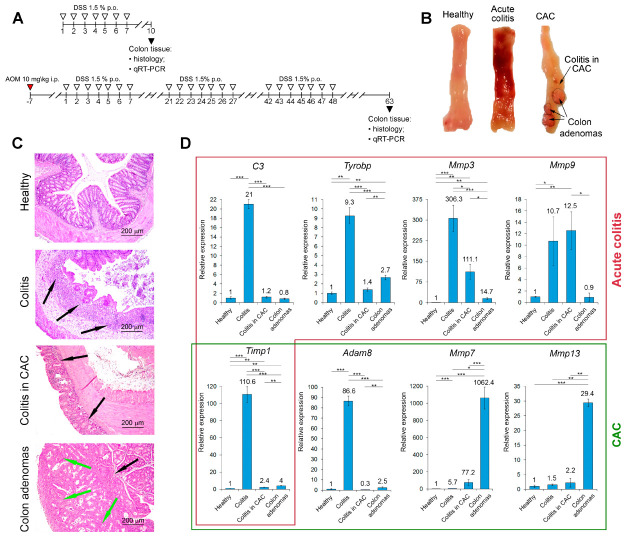
Morphological changes and expression levels of key genes identified by the bioinformatics analysis in the colon tissues of healthy mice and mice with acute colitis and colitis-associated cancer (CAC). (**A**) Experimental setup. Acute colitis was induced by the administration of 2.5% dextran sulfate sodium (DSS) solution in drinking water for 7 days followed by a 3 day recovery (upper panel) (*n* = 10). CAC was induced in mice by single intraperitoneal (i.p.) injection of azoxymethane (AOM) 1 week before DSS administration. Mice were then exposed to 3 consecutive cycles of 1.5% DSS instillations for 7 days followed by 2 weeks recovery (lower panel) (*n* = 10). At the end of the experiment, the colons were collected for subsequent histological analysis and qRT-PCR. (**B**,**C**) Gross morphology (**B**) and histology (**C**) of the colon tissue of healthy mice and mice with acute colitis and CAC. Hematoxylin and eosin staining. Original magnification ×100. Black arrows indicate inflammatory infiltration. Green arrows indicate adenomatous transformation of colon epithelium. (**D**) Expression levels of identified genes in the colon tissues measured by qRT-PCR. HPRT was used as a reference gene. Healthy—healthy colon tissue, colitis—colon tissue with acute colitis, colitis in CAC—colon tissue with chronic colitis adjacent to adenomas. The data are expressed as the mean ± SD (*n* = 6). * *p* < 0.05, ** *p* < 0.01, *** *p* < 0.001.

**Figure 4 ijms-24-04311-f004:**
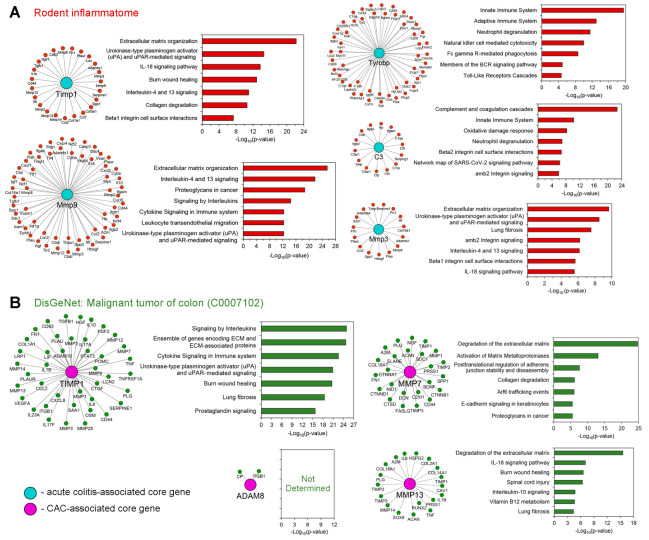
The involvement of the identified acute colitis- and CAC-related hub genes in murine inflammatome (**A**) and the gene network associated with malignant tumor of the colon, retrieved from DisGeNET (**B**). The modules containing the explored genes with their gene partners were extracted from gene networks created using the STRING database and visualized using Cytoscape, according to Section 4.3. Functional analysis of gene modules was performed using the ToppFun tool.

**Figure 5 ijms-24-04311-f005:**
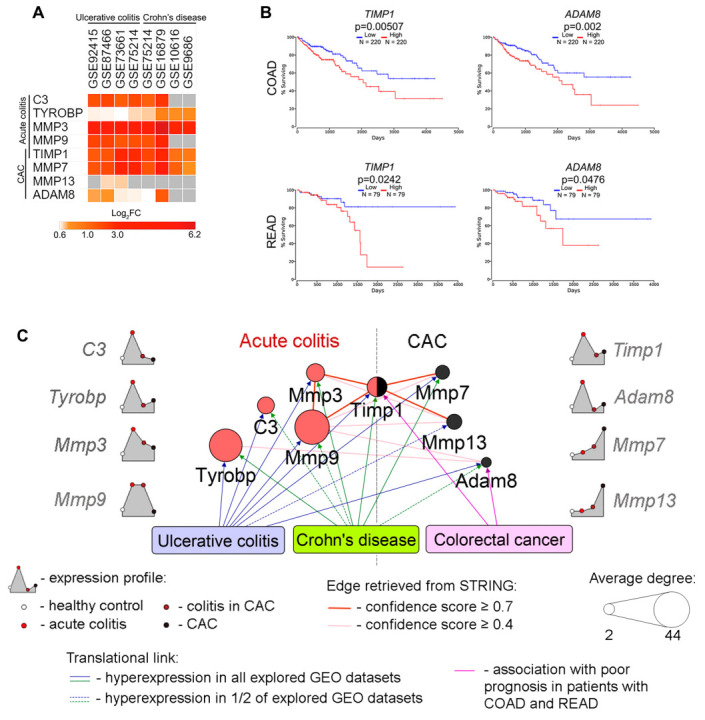
Expression levels of identified DEGs in human patients with inflammatory bowel diseases (ulcerative colitis and Crohn’s disease) and their association with the survival of patients with colon adenocarcinoma (COAD) and rectal adenocarcinoma (READ). (**A**) Heatmap demonstrating the expression levels of identified acute colitis- and CAC-related hub genes in datasets of human patients with ulcerative colitis and Crohn’s disease. The differential analysis of cDNA microarray data was performed according to Section 4.1. The heat map was constructed using Morpheus. Log_2_FC = Log_2_(fold change). In all datasets, biopsy samples were taken from the colon at the sites of active inflammation. (**B**) Survival of patients with COAD and READ depending on the expression levels of identified DEGs (*TIMP1* and *ADAM8*) in the colon tissue. Kaplan–Meier survival curves were constructed using TCGA data according to Section 4.8. (**C**) The final scheme of core genes associated with acute colitis and CAC revealed in the current study, their expression profiles in murine models, and association with IBD and colorectal cancer in humans.

**Table 1 ijms-24-04311-t001:** The GEO microarray datasets used in the study.

Object	Pathology	GEO ID	Murine Strain	Sex	Analyzed Groups	Number of DSS Cycles
Mice	Acute colitis	GSE31106	ICR	m	3 untreated mice, 3 AOM/DSS-treated mice	1
		GSE35609	ICR	f	4 untreated mice, 4 TNBS-treated mice	–
		GSE42768	C57Bl6	f	3 untreated mice, 3 DSS-treated mice	1
		GSE64658	C57Bl6	f	6 untreated mice, 3 AOM/DSS-treated mice	1
		GSE71920	C57Bl6	m	3 untreated mice, 3 DSS-treated mice	1
	CAC	GSE31106	ICR	m	3 untreated mice, 3 AOM/DSS-treated mice	3
		GSE42768	C57Bl6	f	3 untreated mice, 3 DSS-treated mice	3
		GSE56050	Lrg5-lacZ	m	2 untreated mice, 2 AOM/DSS-treated mice	3
		GSE64658	C57Bl6	f	6 untreated mice, 6 AOM/DSS-treated mice	3
Humans	Ulcerative colitis (UC)	GSE73661	–	f, m	12 healthy samples, 67 UC samples	–
		GSE75214	–	f, m	11 healthy samples, 74 UC samples	–
		GSE87466	–	f, m	21 healthy samples, 27 UC samples	–
		GSE92415	–	f, m	21 healthy samples, 87 UC samples	–
	Crohn’s disease (CD)	GSE9686	–	f, m	8 healthy samples, 11 CD samples	–
		GSE10616	–	f, m	26 healthy samples, 18 CD samples	–
		GSE16879	–	f, m	6 healthy samples, 18 CD samples	–
		GSE75214	–	f, m	11 healthy samples, 59 CD samples	–

**Table 2 ijms-24-04311-t002:** Primers used in the study.

Gene	Type	Sequence
*Adam8*	Forward	5′-TATGCAACCACAAGAGGGAG-3′
	Probe	5′-((5,6)-FAM)-TCATCTGATACATCTGCCAGCCGC-3′–BHQ1
	Reverse	5′-ACCAAGACCACAACCACAC-3′
*C3*	Forward	5′-GTTTATTCCTTCATTTCGCCTGG-3′
	Probe	5′-((5,6)-FAM)-ACACCCTGATTGGAGCTAGTGGC-3′–BHQ1
	Reverse	5′-GATGGTTATCTCTTGGGTCACC-3′
*Timp1*	Forward	5′-CTCAAAGACCTATAGTGCTGGC-3′
	Probe	5′-((5,6)-FAM)-ACTCACTGTTTGTGGACGGATCAGG-3′–BHQ1
	Reverse	5′-CAAAGTGACGGCTCTGGTAG-3′
*Tyrobp*	Forward	5′-GGTGACTTGGTGTTGACTCTG-3′
	Probe	5′-((5,6)-FAM)-CCTTCCGCTGTCCCTTGACCTC-3′–BHQ1
	Reverse	5′-GACCCTGAAGCTCCTGATAAG-3′
*Mmp3*	Forward	5′-TGCATATGAGGTTACTAACAGAGAC-3′
	Probe	((5,6)-FAM)-5′-AATCAGTTCTGGGCTATACGAGGGC-3′-BHQ1
	Reverse	5′-CAGGGTGTGAATGCTTTTAGG-3′
*Mmp7*	Forward	5′-CATAATTGGCTTCGCAAGGAG-3′
	Probe	((5,6)-FAM)-5′-TACTGGACTGATGGTGAGGACGCA-3′-BHQ1
	Reverse	5′-CAAATTCATGGGTGGCAGC-3′
*Mmp9*	Forward	5′-ACCTGAAAACCTCCAACCTC-3′
	Probe	((5,6)-FAM)-5′-TAGCGGTACAAGTATGCCTCTGCC-3′-BHQ1
	Reverse	5′-TCGAATGGCCTTTAGTGTCTG-3′
*Mmp13*	Forward	5′-GATTATCCCCGCCTCATAGAAG-3′
	Probe	((5,6)-FAM)-5′-CAGCATCTACTTTGTTGCCAATTCCAGG-3′-BHQ1
	Reverse	5′-CCCACCCCATACATCTGAAAG-3′

## Data Availability

The data presented in this study are openly available on the Gene Expression Omnibus database. Reference numbers: GSE31106, GSE35609, GSE42768, GSE64658, GSE71920, GSE31106, GSE42768, GSE56050, GSE64658, GSE73661, GSE75214, GSE87466, GSE92415, GSE9686, GSE10616, GSE16879, and GSE75214.

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
