# Peer review of "Identification of Novel Core Genes Involved in Malignant Transformation of Inflamed Colon Tissue Using a Computational Biology Approach and Verification in Murine Models"

_ijms, 2023, doi:10.3390/ijms24054311_

Round 1
Reviewer 1 Report
The authors aimed to explore the molecular mechanism of colitis-driven tumorigenesis in mice. They, by using a comprehensive bioinformatics analysis of multiple 13 transcriptomics datasets from the colon tissue of mice with acute colitis and colitis-associated cancer (CAC) revealed a set of key overexpressed genes involved in the regulation of colitis (C3, 17 Tyrobp, Mmp3, Mmp9, Timp1) and CAC (Timp1, Adam8, Mmp7, Mmp13), also interconnected between each other. Further validation of obtained data in murine models of dextran sulfate sodium (DSS)-induced colitis and azoxymethane/DSS-stimulated CAC fully confirmed the association of revealed hub genes with inflammatory and malignant lesions of colon tissue and demonstrated that genes encoding matrix metalloproteinases (acute colitis: Mmp3, Mmp9; CAC: Mmp7, Mmp13) can be used as a novel prognostic signature for colorectal neoplasia in IBD. Finally, they evaluated the transcriptomics profiles of colon tissue from patients with UC and CD (GEO datasets), with colorectal cancer (Cancer Genome Atlas-TCGA), as well as the protein level of TIMP1 and ADAM8 in the colon tissue of colorectal cancer patients (Human Protein Atlas data). The identified core genes are overexpressed in colon tissue of patients with IBD (all explored genes) and highly aggressive colorectal cancer (TIMP1, ADAM8) that confirms the interest in studying these genes within the framework of intestinal pathologies in humans.
Reviewer’s points
- Title. Since most of the bioinformatics analyzes carried out, as well as the in vivo experiments were done on the mouse model, it would be appropriate to add to the title, at its end: “in murine models”.
- Mat. and Met. Indicate the number and sex of mice used in each in vivo experiment.
- Discussion. A different expression of TIMP1 between males and females has been reported in the literature (Chris D. Hermann et al. J. Exp. Med. 2021, Vol.218;Nò.11; e20210911), the authors should indicate in the discussion whether this was taken into account in the evaluation of the obtained results.
Author Response
Dear Reviewer #1,
We are very grateful to you for the valuable comments that helped us improve the manuscript. We revised the manuscript according to your comments, and, please, let us respond to your remarks.
- Title. Since most of the bioinformatics analyzes carried out, as well as the in vivo experiments were done on the mouse model, it would be appropriate to add to the title, at its end: “in murine models”.
Authors: Corrected. The title of the manuscript was replaced with “Identification of novel core genes involved in malignant transformation of inflamed colon tissue using a computational biology approach and verification in murine models”. Please, see lines 2-4.
- Mat. and Met. Indicate the number and sex of mice used in each in vivo experiment.
Authors: Corrected. Thanks a lot for the careful analysis of our manuscript! It was our disappointing omission to not indicate this information on animal experiments. The number of mice used in the validation study (n = 10 (induction of colitis and CAC), n = 6 (qRT-PCR)) were added in Section 4.5 (please, see lines 589, 591) and Section 4.7 (please, see lines 620-621). Moreover, the number of mice was also added in the legend to Figure 3 (please, see lines 281, 283, 291). The sex of the mice (female) is shown in the line 581.
- Discussion. A different expression of TIMP1 between males and females has been reported in the literature (Chris D. Hermann et al. J. Exp. Med. 2021, Vol.218;Nò.11; e20210911), the authors should indicate in the discussion whether this was taken into account in the evaluation of the obtained results.
Authors: Corrected. Since our bioinformatics study had been performed using GEO datasets belonging to mice of both sexes and the validation of in silico results was performed in female mice only, sex disparities in the expression of core genes did not considered during the analysis of obtained data. To indicate this fact, the phrase describing the findings of Hermann et al. and an interest to further exploration of Timp1 expression in CAC mice of both sexes was added in the Discussion (please, see lines 496-498) and in novel section “Limitations of the study” (please, see lines 513-516). Besides, the information about sex of mice with colitis and CAC used for cDNA microarray analysis was added in Table 1 (please, see p. 15).
We hope that corrected version of the manuscript will be acceptable for publication in the International Journal of Molecular Sciences.
Sincerely,
On behalf of all authors,
Dr. Andrey Markov

Reviewer 2 Report
The paper by Markov et al. reports very interesting findings that showed translational bridge interconnecting of listed colitis/CAC-associated core genes with the pathogenesis of ulcerative colitis, Crohn’s disease, and colorectal cancer in humans. The authors found a set of key overexpressed genes involved in the regulation of colitis (C3,Tyrobp, Mmp3, Mmp9, Timp1) and CAC (Timp1, Adam8, Mmp7, Mmp13) occupied hub positions within explored colitis- and CAC-related regulomes using performed intersection of differentially expressed genes (DEGs), their functional annotation, reconstruction, and topology analysis of gene association networks combined with the text mining approach. In addition, they validated the obtained data using the murine models of dextran sulfate sodium (DSS)-induced colitis and azoxymethane/DSS-stimulated CAC. Finally, the authors strengthened that the findings serve both as promising molecular markers and therapeutic targets to control IBD and IBD-associated colorectal neoplasia. I think the findings are of interest and the manuscript is well written. Experiments were carefully conducted. I have just a few comments for publication.
1) Text and Figure 3 legend: “CAC” is sometimes confused. “CAC” must be “colitis-associated cancer" but not “colon adenoma” Please check again.
2) Important papers (PMID: 14611673 and PMID: 17506908) for the murine model of CAC and gene expression are not cited. Please cite and discuss, if possible.
Author Response
Dear Reviewer #2,
We are very grateful to you for your careful review of our manuscript. We revised the manuscript according to your comments, and, please, let us respond to them.
- Text and Figure 3 legend: “CAC” is sometimes confused. “CAC” must be “colitis-associated cancer" but not “colon adenoma” Please check again.
Authors: Corrected. Thank you for the identification of this confusing misprint. Figure 3B, C, D and the text in Section 2.3.2 were corrected, according to this remark (incorrect CAC was replaced with “Colon adenomas”; please see Figure 3B, C, D (p. 8), lines 290 (incorrect CAC was deleted), and 324.
- Important papers (PMID: 14611673 and PMID: 17506908) for the murine model of CAC and gene expression are not cited. Please cite and discuss, if possible.
Authors: Corrected. Indeed, during preparing the manuscript, we concentrated our attention only on recent publications in the field of colitis, CAC, and their transcriptomics analysis; however, important pioneering reports describing methods for obtaining of murine models of our interest did not be included in the text. This shortcoming was corrected: the works of Suzuki et al. and Tanaka et al. were added to the Introduction (please, see line 66, reference 12) and Section 4.5 (please, see lines 592-593).
We hope that this version of the manuscript will be acceptable for publication.
Thank you very much!
Sincerely,
On behalf of all authors,
Dr. Andrey Markov

Reviewer 3 Report
This submitted manuscript aims to investigate the molecular mechanisms of colitis-associated cancer (CAC) using bioinformatic analysis of publicly available datasets, and showed that overexpression of a list of genes may be involved in the tumorigenesis of CAC. Among them, TIMP1 and ADAM8 might serve as prognostic predictors in patients with colon adenocarcinoma (COAD) or rectal adenocarcinoma (READ). However, this is unacceptable because the study design, particularly the translational study of human data, is seriously flawed and invalidates the conclusions. The concerns raised are listed below:
Comments to author:
1. Please emphasize this is an animal-based study in the abstract.
2. Please explain why the authors used only female mice in this study?
3. Please describe how many mice were used for validation?
4. The legend for Figure 3D is absent, or I guess it was miss-labeled as (C). Please check.
5. What does the error bars mean in Figure 3D? Please describe in the related legend.
6. The section 4.5 and 4.6 can be combined.
7. It is unclear whether Figure 4 and 5 are from this study or another study. If it is from this study, why were they only mentioned in the discussion but not in the result section?
8. Figure 5B, which are from the TCGA COAD and READ cohorts, but not pure UC or CD-associated CAC patients. Therefore, this result may not provide further evidence to support the specific signatures in CAC patients as mentioned in the conclusion.
9. The legend for Figure 5C states that the protein levels were obtained from The Human Protein Atlas. However, to my understanding, The Human Protein Atlas provides only mRNA levels for prognostic correlation, not protein levels. The protein levels predominantly derived from the immunohistochemical imaging evidence in a small number of patients, impossible to provide prognosis in such a large number of cases. Please explain.
Author Response
Dear Reviewer #3,
We are genuinely thankful to you for your careful analysis of our manuscript and highly valuable remarks. We revised the manuscript according to your comments, and, please, let us respond to them.
TIMP1 and ADAM8 might serve as prognostic predictors in patients with colon adenocarcinoma (COAD) or rectal adenocarcinoma (READ). However, this is unacceptable because the study design, particularly the translational study of human data, is seriously flawed and invalidates the conclusions.
Authors: Corrected. Dear Reviewer #2, thank you for this important criticism. Indeed, a thorough and non-prejudiced re-analysis of the text of the manuscript performed by us revealed that the conclusions related to our preliminary translational study may mislead readers. To fix this problem, according to your comments, the following important corrections were introduced into the manuscript:
- The novel section named “Limitations of the study” was added after the Discussion (please, see lines 511-524). Among other, this section contains the following phrase, describing the linkage of our data with the pathogenesis of colon diseases in humans: “considering that our findings are predominantly animal-based, to more clearly elucidate how closely and even whether identified core genes are involved in the regulation of intestinal pathologies in humans, revealed translational bridge needs further large-scale verification study, using clinical samples of patients with UC, CD, and UC/CD-associated colorectal cancer”.
- Throughout the manuscript, we emphasize that (i) our findings are animal-based (please, see lines 4, 13, 516, 654), (ii) identified association of core genes with human pathologies is only presumptive (please, see lines 493, 658 (Conclusion)), and (iii) human-related study was performed using only publicly available transcriptomics data (please, see lines 24-25 (Abstract)).
- Please emphasize this is an animal-based study in the abstract.
Authors: Corrected. Please, see line 13. Moreover, this phrase was introduced in the Limitations of the study and Discussion (please, see lines 516 and 654, respectively).
- Please explain why the authors used only female mice in this study?
Authors: Indeed, the current rules concerning the studies on laboratory animals demand the usage of both sexes [1] urging the researchers to account for sex as a biological variable in preclinical studies. However, given our study is a research one (not a preclinical work), in which murine experiments were used to confirm the obtained in silico data. Moreover, a relatively recent study of Dr. Beery demonstrates, that the usage of female rodents does not increase the variability in studies [2]. Thus, we saw no justifiable need to include mice of both sexes in the in vivo part of our study. According to this comment, the phrase describing the need for further verification of obtained data in mice of both sexes was added to the section Limitations of the study (please, see lines 512-515).
- Clayton, J.A. Studying both sexes: A guiding principle for biomedicine. FASEB J. 2016, 30, 519–524, doi:10.1096/fj.15-279554.
- Beery, A.K. Inclusion of females does not increase variability in rodent research studies. Curr. Opin. Behav. Sci. 2018, 23, 143, doi:10.1016/J.COBEHA.2018.06.016.
- Please describe how many mice were used for validation?
Authors: Corrected. We sincerely apologize for this omission. The number of mice used in the validation study (n = 10 (induction of colitis/CAC), n = 6 (qRT-PCR)) was added to section 4.5 (please, see lines 589, 591) and section 4.7 (please, see lines 620-621). Moreover, the number of mice was also added to the legend of Figure 3 (please, see lines 281, 283, 291).
- The legend for Figure 3D is absent, or I guess it was miss-labeled as (C). Please check.
Authors: Corrected. Please, see line 288. You are right, it was a misprint.
- What does the error bars mean in Figure 3D? Please describe in the related legend.
Authors: Corrected. Please, see lines 290-291.
- The section 4.5 and 4.6 can be combined.
Authors: Corrected. Please, see lines 580-598.
- It is unclear whether Figure 4 and 5 are from this study or another study. If it is from this study, why were they only mentioned in the discussion but not in the result section?
Authors: The results depicted in Figures 4 and 5 were obtained by us in the current study. The description of these results in the Discussion aimed to show the interconnection of our findings with the pathogenesis of colitis and CAC not only by the analysis of published literature but also independently using the in silico approach (for instance, a phrase describing this approach can be found in line 471). In order to more clearly indicate that these results are from the current study, the references on the description of used methods were added to the legends of Figure 4 (please, see lines 397-398) and Figure 5 (please, see lines 530-531, 535).
- Figure 5B, which are from the TCGA COAD and READ cohorts, but not pure UC or CD-associated CAC patients. Therefore, this result may not provide further evidence to support the specific signatures in CAC patients as mentioned in the conclusion.
Authors: Corrected. Indeed, performed TCGA analysis did not consider the ratio of UC and CD-associated CAC patients in COAD and READ cohorts. To not mislead the readers, the phrase describing this limitation was added to the text of the manuscript (please, see lines 492-496).
- The legend for Figure 5C states that the protein levels were obtained from The Human Protein Atlas. However, to my understanding, The Human Protein Atlas provides only mRNA levels for prognostic correlation, not protein levels. The protein levels predominantly derived from the immunohistochemical imaging evidence in a small number of patients, impossible to provide prognosis in such a large number of cases. Please explain.
Authors: Corrected. Dear Reviewer #3, thanks a lot for the identification of this serious mistake! Indeed, the survival analysis data deposited in The Human Protein Atlas is based on RNA-seq data (FPKM). We sincerely apologize for this negligence. All information describing the protein levels of TIMP1 and ADAM8 was deleted from the manuscript (Please, see corrected Figure 5).
We hope that this version of the manuscript will be acceptable for publication.
Thank you very much!
Sincerely,
On behalf of all authors,
Dr. Andrey Markov

Round 2
Reviewer 3 Report
The authors have completely answered the questions raised by reviewer. It is more clear and is now acceptable.